# Heavy Metal Concentration Estimation for Different Farmland Soils Based on Projection Pursuit and LightGBM with Hyperspectral Images

**DOI:** 10.3390/s24103251

**Published:** 2024-05-20

**Authors:** Nan Lin, Xiaofan Shao, Huizhi Wu, Ranzhe Jiang, Menghong Wu

**Affiliations:** 1College of Surveying and Exploration Engineering, Jilin Jianzhu University, Changchun 130118, China; linnan@jlju.edu.cn (N.L.); shaoxiaofan@student.jlju.edu.cn (X.S.); wumenghong@jlju.edu.cn (M.W.); 2Jilin Province Natural Resources Remote Sensing Information Technology Innovation Laboratory, Changchun 130118, China; 3Henan Academy of Geology, Zhengzhou 450016, China; 4College of Biological and Agricultural Engineering, Jilin University, Changchun 130012, China; jiangrz23@mails.jlu.edu.cn; 5College of Resource and Environmental Science, Jilin Agricultural University, Changchun 130118, China

**Keywords:** hyperspectral image, soil heavy metals, local regression estimation, projection pursuit, LightGBM

## Abstract

Heavy metal pollution in farmland soil threatens soil environmental quality. It is an important task to quickly grasp the status of heavy metal pollution in farmland soil in a region. Hyperspectral remote sensing technology has been widely used in soil heavy metal concentration monitoring. How to improve the accuracy and reliability of its estimation model is a hot topic. This study analyzed 440 soil samples from Sihe Town and the surrounding agricultural areas in Yushu City, Jilin Province. Considering the differences between different types of soils, a local regression model of heavy metal concentrations (As and Cu) was established based on projection pursuit (PP) and light gradient boosting machine (LightGBM) algorithms. Based on the estimations, a spatial distribution map of soil heavy metals in the region was drawn. The findings of this study showed that considering the differences between different soils to construct a local regression estimation model of soil heavy metal concentration improved the estimation accuracy. Specifically, the relative percent difference (*RPD*) of As and Cu element estimations in black soil increased the most, by 0.30 and 0.26, respectively. The regional spatial distribution map of heavy metal concentration derived from local regression showed high spatial variability. The number of characteristic bands screened by the PP method accounted for 10–13% of the total spectral bands, effectively reducing the model complexity. Compared with the traditional machine model, the LightGBM model showed better estimation ability, and the highest determination coefficients (*R*^2^) of different soil validation sets reached 0.73 (As) and 0.75 (Cu), respectively. In this study, the constructed PP–LightGBM estimation model takes into account the differences in soil types, which effectively improves the accuracy and reliability of hyperspectral image estimation of soil heavy metal concentration and provides a reference for drawing large-scale spatial distributions of heavy metals from hyperspectral images and mastering soil environmental quality.

## 1. Introduction

Soil is a complex, dynamic, and multicomponent natural system on Earth’s surface [1] that provides necessary material resources for the survival of plants, animals, and humans, and maintaining the balance of soil structure, composition, and biophysicochemical properties is vital for maintaining soil functions [2]. However, intensive human activities, such as the irrational use of pesticides, livestock wastewater discharge, and coal combustion [3], have caused excessive heavy metal concentration in soil and threatened soil environmental quality [4]. From the standpoint of agricultural productivity, excessive heavy metals reduce soil fertility, thus affecting crop output and the growth cycle [5]. Meanwhile, heavy metals accumulated in soil are absorbed by crops and then affect animal and human health via the food chain [3,6]. For example, excessive arsenic (As) in soil inhibits seed germination and reduces the seed setting rate [7], excessive copper (Cu) inhibits growth and triggers antioxidant responses in crops such as rice and corn [8], and heavy metals entering the human body may exert toxic effects on various organs [9]. In recent years, As and Cu released by human activities in various regions have been continuously superimposed on the background of soil As and Cu, thus forming regional differences in soil As and Cu pollution [10,11]. At present, it has become an important task for soil environmental quality monitoring to quickly grasp the regional soil environmental status, especially whether the accumulation of heavy metals in farmland soil exceeds the standard [12]. However, traditional soil heavy metal pollution monitoring relies mainly on geochemical methods [13,14], which have deficiencies, such as limited monitoring ranges, high time consumption, and high cost [15]. In order to offer solid data support for the thorough treatment and environmental remediation of heavy metals in soil, quick and reliable technologies are desperately needed for monitoring and grasping heavy metal concentration and spatial distribution in agricultural soil.

With rich spectral information, broad spatial coverage, and long time series, hyperspectral imaging has emerged as a new non-contact earth observation technology that can quickly and affordably estimate the regional spatial distribution of heavy metals in soils and monitor their changes [16]. Soil spectral reflectance comprehensively reflects the soil physicochemical properties [17], where the main soil components, such as organic matter, iron and manganese oxides, and clay minerals, significantly correlate with soil spectra [18]. Heavy metals have strong adsorption and occurrence relationships with these major soil fractions and, therefore, have significant correlations with soil spectra [4,19]. Such correlations provide the basis for developing statistical models to estimate soil heavy metal concentrations [20]. Numerous studies have shown the feasibility of using hyperspectral imaging to construct soil heavy metal concentration estimation models [5,21,22]. Thus, how to enhance the accuracy and dependability of soil heavy metal concentration estimation models with hyperspectral imaging has received close attention.

Currently, the high-dimensional redundancy of spectrum data, the advanced nature of the model, and the geographical heterogeneity of the soil environment are the key factors affecting the estimation accuracy of soil heavy metal concentration. Hyperspectral imaging records soil spectral information on continuous and compact bands with many spectral channels, extensive data, and rich information. Due to their large quantities, different bands may contain much similar information, i.e., information redundancy, which seriously affects the stability and accuracy of the estimation models [23]. In response to this problem, feature selection algorithms such as the correlation coefficient thresholding method, successive projection algorithm (SPA), uninformative variable elimination (UVE), and competitive adaptive reweighted sampling (CARS) have been adopted to select suitable spectral feature bands, thus effectively reducing the data redundancy and accelerating the training speed [24,25,26]. With limited training samples, however, the above methods face issues such as inaccurate parameter estimation and incomplete feature analysis of high-dimensional data, causing the selected feature bands to fall below expectations. The projection pursuit (PP) method is widely used in the dimensionality reduction of high-dimensional data, which improves the parameter estimation accuracy by optimizing the projection index. Under the premise of retaining the main band information, the high-dimensional data is reduced, and the spectral eigenbands are screened out by using low-dimensional space computation [27,28], which effectively solves the high autocorrelation between the spectral bands and improves the model estimation accuracy. Regarding estimation model algorithm selection, machine learning models have shown good ability in soil heavy metal concentration estimation with excellent feature mining and data fitting capabilities [20,29]. For example, support vector regression (SVR) [30,31], extreme learning machines (ELM) [32], and random forests (RF) [33,34,35]. However, traditional machine learning algorithms are limited by training data and prone to problems such as complicated parameter tuning, slow training speed, and model overfitting during model estimation, resulting in poor estimation accuracy [36]. In recent years, ensemble learning algorithms have demonstrated higher accuracy and generalization ability than machine learning algorithms in cases of training data scarcity [37,38]. LightGBM is an ensemble learning algorithm with high regression estimation accuracy. The samples with tiny gradients are filtered using the unilateral gradient algorithm during training, and the ergodic samples are transformed into ergodic histograms using the histogram optimization algorithm. The leaf-wise algorithm’s splitting strategy greatly reduces the time complexity and calculation steps while improving the algorithm’s training speed. The LightGBM framework adopts the gain-first splitting strategy, which requires fewer parameter adjustments and prevents model overfitting and the problem of model accuracy degradation caused by changes in data distribution [39,40].

In addition, soil heterogeneity also affects the accuracy of hyperspectral imaging-based estimation models to some degree. The significant spatial differences in soil heterogeneity lead to sample imbalances [41] and, in turn, result in the different heavy metal enrichment levels of different soils [42]. The different concentrations of spectral active components in different soils [43] increase model complexity to a certain extent, thus reducing the estimation accuracy and reliability of the model. Bao et al. [44] confirmed the necessity of soil classification for model estimation. Xiao et al. [45] extracted spectral bands sensitive to soil element concentrations based on different soils, greatly improving the estimation efficiency. Therefore, extracting input variables by considering the differences in different soils during model construction and constructing a local regression estimation model of soil heavy metal concentration can improve the model estimation accuracy.

This study proposes a method of soil heavy metal concentration estimation based on hyper-spectral imaging. ZY1-02D remote sensing images were used as the data source, a local regression model was constructed by distinguishing soil types to improve the accuracy of hyperspectral imaging in estimating soil heavy metal concentration with limited samples. Sihe Town and its surrounding agricultural area in Yushu City, Jilin Province, China, were selected as the study area. A total of 440 soil samples covering four distinct soil types were collected from the study area. The As and Cu concentration data were used to investigate the spectral properties of various soils. The PP algorithm was used to screen the sensitive bands. The LightGBM framework was used to create the soil heavy metal estimation model and draw the concentration distribution map in the study area, thus determining the soil heavy metal accumulation degree in the study area and analyzing the potential influencing factors. The objectives of this study are as follows: (1) explore the effect of local regression models of different soil types on the estimation accuracy of soil heavy metal concentration; (2) combining PP and the LightGBM algorithm to construct the PP–LightGBM estimation model of farmland soil heavy metal concentration; (3) mapping the spatial distribution of heavy metal concentration. Based on the spatial distribution map, the spatial distribution characteristics of heavy metal accumulation in different soil types were analyzed.

## 2. Materials and Methods

### 2.1. Study Area

The study area was in Sihe Town and the surrounding farming area (126°75′–126°93′ E, 44°90′–45°02′ N; covering 166.033 ${km}^{2}$) in the northeast of Yushu City, Jilin Province, China (Figure 1). The area has an average annual temperature of 5.3 °C, with a moderate continental monsoon climate characterized by long and severely cold winters with thick snow cover and short and warm summers with abundant precipitation. The annual average precipitation is 536.4 mm. The terrain in the study area has slight slope undulations, with an average elevation of 186 m. The area has abundant natural resources, sufficient surface water, and various soil types, including black soil, white paste soil, white paste black soil, and meadow soil, which are suitable for crop development. The main crops in the area are corn, rice, and soybeans, and it is an important black soil planting area and grain production base in China.

### 2.2. Datasets

#### 2.2.1. Soil Sample Collection and Analysis

This study employed the Chinese soil classification standards and high-resolution remote sensing images to determine the soil distribution in the region, and the sampling points and sampling routes were determined according to the soil types to ensure even coverage of the study area. Soil samples were collected in mid-April 2022, during the plowing period, and no crop or weed grew on the soil surface, which was convenient for sampling. In compliance with the soil sampling point design, 440 samples were collected. The sampling point distribution is depicted in Figure 1b. In order to eliminate the influence of the mixed pixels at the sampling point on the follow-up study, during the sampling process, the distance between the location of the sampling point and the surrounding ground objects is more than 100 m to ensure that the pixels of the sampling point are pure soil pixels. To ensure the accuracy of the experimental results, the five-point sampling method (Q, $A_{1}$, $A_{2}$, $A_{3}$, and $A_{4}$) was used (Figure 2). Taking the position of each sampling point (Q) as the center, samples were collected in the range of 3 m × 3 m around it. At each sampling point position, 5 soil samples were collected from a 15 cm sampling depth. After the samples were well combined and transferred into the sample bag, a handheld GPS finder (Garmin GPSMAP60CSX, Shanghai, China) was used to record the altitude at point Q, the collection time, and the latitude and longitude coordinates. After collection, all samples were dried in an oven at 60 °C for 48 h, and the stone debris was removed. After sieving through a 0.15 mm screen, the heavy metal concentration was determined. X-ray fluorescence spectrometry (Axios X-ray fluorescence spectrometer, Netherlands) was conducted to determine the levels of two heavy metals, As and Cu, in the soil according to soil agrochemical analysis methods. The determination process strictly followed the requirements of Chinese land quality geochemical assessment specifications. Each sample was repeatedly processed, and the relative standard deviation (RSD) of multiple determination results was calculated as precision. With an accuracy and precision qualification rate above 98%, the results could be used in subsequent analyses.

#### 2.2.2. Remote Sensing Data Preprocessing

The ZY1-02D satellite, launched on 12 September 2019, is the first hyperspectral operational satellite for civilian use in China. Equipped with a 9-band multispectral camera and a 166-band hyperspectral camera, it provides 2500 nm panchromatic, 10 m multispectral, and 30 m hyperspectral image data [46]. In order to ensure the synchronicity between the remote sensing image generation time and the ground experiment, the ZY1-02D satellite images generated on 26 April 2022, were selected according to the sampling time. The cloud coverage of the study area in the selected images was below 5%. Because the spectral channels in some spectral ranges contained much noise or were affected by atmospheric water vapor absorption, the imaging quality of the spectral data was insufficient. Therefore, these spectral channels were screened and eliminated, and the processed hyperspectral images contained 145 bands. To provide a more reliable data source for subsequent hyperspectral imaging-based data analyses, the pixel DN values were converted to apparent reflectance data through radiometric calibration. Meanwhile, atmospheric corrections on the image were performed using the FLAASH module (ENVI 5.3) to eliminate radiometric errors caused by atmospheric molecules and aerosol scattering, thus improving the quality and authenticity of the pixel spectral data [47]. The agriculture boundary was extracted to increase the accuracy of the heavy metal concentration in the research area (Figure 3). The random forest (RF) algorithm was used for hyperspectral image classification supervision, farmland boundary extraction, and non-soil information removal (such as roads and buildings). According to the results, the extracted farmland soil range has clear boundaries with non-agricultural land pixels such as roads and construction land, and the patches are relatively complete, which can be used for subsequent research.

### 2.3. Methods

To effectively improve the estimation accuracy of soil heavy metal concentrations from hyperspectral images, this experiment constructed a method for estimating heavy metal concentrations in soil from hyperspectral images based on considering different soil types (Figure 4). Using the soil type distribution map of the study area and based on the detailed analysis of soil type distribution in the study area, soil samples were collected, the heavy metal concentrations were measured during the collection, and the soil sample data were divided according to the soil type. Based on the sampling time, hyperspectral images of the study area were acquired and preprocessed, and the spectral curve was extracted from the sampling point coordinates. To compare and analyze the spectral feature variations of the various soil types, the spectral curves of the different types of soil pixels were analyzed based on their heavy metal concentrations. Under the analysis results, the PP algorithm was utilized to extract the distinctive spectral bands of soil heavy metals in the samples of various soil types and the overall sample. Finally, a soil heavy metal concentration estimation model based on the characteristic spectral bands was developed using the LightGBM framework to derive the soil heavy metal concentration distribution map. The specific process is as follows.

#### Projection Pursuit Method

PP is an unsupervised linear dimensionality reduction method applied to high-dimensional data [48]. The high-dimensional data are projected onto low-dimensional (1 to 3) visual subspaces to derive projections reflecting the original high-dimensional data structure or features (interesting projections), which are analyzed and studied to understand the original high-dimensional data [49]. The basic principles are as follows. (1) The variance is selected as the projection index, and the genetic algorithm is employed to find the projection direction with the optimal projection index $a_{1}$. (2) After obtaining the projection direction $a_{1}$, the data projection value *X* in this direction $a_{1}^{T}X$ becomes the first data projection structure of interest, i.e., principal component 1. (3) The data are projected onto a subspace orthogonal to $a_{1}$, and the projection finding process is repeated for the projected data. (4) Let the *j* principal component be *X*(*j*), and the projection value after the PP is $a_{j}^{T}X$, where $a_{j}$ is the characteristic root corresponding to $\lambda_{j}$; then, the eigenvalues of the principal components and the variances of the corresponding principal components equal $Var(X_{j})=\lambda_{j}$, j=1,2……n. (5) Each principal component $X_{k}$ is calculated. The contribution rate of $\lambda_{k}/\sum_{1}^{n}\lambda_{j}$ is set to 90%. When the cumulative contribution rate of the principal components reaches 90%, the optimization process ends. The corresponding principal components are the data structure characteristics of interest obtained by retaining the optimum. These principal components are the model input data with important band information.

### 2.4. Model Evaluation Methods

#### 2.4.1. Light Gradient Boosting Machine

LightGBM is a fast, high-performance boosting ensemble learning framework based on the decision tree algorithm with fewer parameters [50]. Compared with other boosting algorithms, LightGBM has a faster training speed, fewer parameter choices, and a higher accuracy rate [51]. The specific calculation method is as follows:

Given a dataset $B=\left\{\left(X_{i},y_{i}\right)\right\}\left(X_{i}=H^{m},y_{i}=H\right)$, n is the sample size, and m is the number of features. $X_{i}=\left[F_{i},x_{i_{1}},x_{i_{2}},\ldots ,x_{i_{s}}\right]$ is the characteristic parameter of the i sample, $F=\left\{F_{1},F_{2},\ldots ,F_{n}\right\}$ is the heavy metal element concentration, and the prediction results of element concentration is:(1)$$y_{i}=\sum_{g=1}^{G}f_{g}\left(X_{i}\right),f_{g}=\eta ,$$
where $\eta =\left\{f\left(X\right)=\theta_{p\left(x\right)}\right\}\left(p:H^{m}YT,\theta =H^{T}\right)$ represents the regression tree data space; p represents a tree structure that maps a sample to the corresponding leaf node T; and each $f_{g}$ represents an independent tree structure and leaf weights θ.

Then, the objective function is:(2)$$L=\sum_{i}\left(y_{i},y_{i}^{\prime }\right)+\sum_{g}\Omega \left(f_{g}\right),$$
where
(3)$$l\left(y_{i},y_{i}^{\prime }\right)={\left(y_{i},y_{i}^{\prime }\right)}^{2},$$
represents the loss function and can evaluate the error between the predicted value $y_{i}^{\prime }$ and target value $y_{i}$;
(4)$$\Omega \left(f_{g}\right)=\eta T+\frac{1}{2}\lambda {\left|\theta \right|}^{2},$$
represents the penalty term of the model and is adjusted by η and λ. Parameters can affect the fitting degree of the model. Table 1 lists the main training parameters of the model. Among them, num_leaves is the main parameter to control the complexity of the decision tree model, which can limit the maximum depth of the decision tree and prevent the model from overfitting. As the learning rate of the model, the learning_rate is generally set between 0.05 and 0.1. Selecting a relatively small learning rate can improve the model’s performance. The early_stopping_rounds parameter indicates that the training will stop when the metric of the validation set is no longer improved after several cycles. In this study, the early_stopping_rounds parameter is set to 50 based on multiple simulations.

#### 2.4.2. Model Evaluation and Accuracy Verification

To assess the model’s stability and estimation performance, the heavy metal concentration estimation findings were compared across several soil types. This study used the coefficient of determination ($R^{2}$), root mean square error (*RMSE*) and relative analytical error (*RPD*) to evaluate the predictive ability of the model. The formula is as follows:(5)$$R^{2}=1-\sum_{i=1}^{n}{\left(Q_{i}-M_{i}\right)}^{2}/\sum_{i=1}^{n}{\left(Q_{i}-\bar{Q}\right)}^{2},$$
(6)$$RMSE=\sqrt{\sum_{i=1}^{n}{\left(Q_{i}-M_{i}\right)}^{2}/n}$$
(7)RPD=Std/RMSE

In the formula, n is the number of samples involved in the modeling, $Q_{i}$ and $M_{i}$ are the measured true values and model estimates, respectively, $\bar{Q}$ is the average value of all sample measurements, and Std is the standard deviation of the sample. The relative percent difference RPD range can be divided into three grades from small to large. When RPD is less than 1.4, the model estimation ability is low; when RPD is between 1.4 and 2, the model is useful for estimating soil heavy metal concentrations. When RPD is above 2, the estimation ability of the model is strong.

## 3. Results

### 3.1. Soil Heavy Metal Concentration Analysis

The study area has four primary soil types: meadow soil, albic black soil, black soil, and albic soil, covering 21.05%, 17.01%, 32.89%, and 29.05% of the study area, respectively. The sampling points were divided according to the soil types. The numbers of black soil, albic black soil, albic soil, and meadow soil samples were 122, 95, 116, and 107, respectively. The heavy metal concentration information of the soil samples is presented in Table 2. It can be observed that the heavy metal concentrations increased significantly compared with the background value of the study area, with As concentrations ranging from 7.24 mg/kg to 13.53 mg/g and Cu concentrations ranging from 16.51 mg/kg to 29.51 mg/kg. By comparing the heavy metal concentrations of the different soils, it was found that the mean cumulative As concentration was the highest in meadow soil (mean = 9.96 mg/kg), and the mean cumulative Cu concentration was the highest in black soil (mean = 22.17 mg/kg).

### 3.2. Spectral Characteristic Analysis of the Different Soils

As a trace element, heavy metals in soil have a weak influence on soil spectrum. However, the high correlation components with soil spectra such as clay minerals and organic matter have strong adsorption. Therefore, the relations between heavy metal concentration and soil spectral were established [52]. The different compositions and physicochemical properties of different soils affect the reflectance and absorption spectral characteristics, resulting in different soil spectral measurements. To compare and analyze the spectral characteristic differences of different soils and their changes with heavy metal concentrations. The extracted image pixel spectra were divided into black soil, albic black soil, albic soil, and meadow soil spectra, which were subdivided into five groups from low to high according to the As and Cu concentrations. The hyperspectral data and the mean heavy metal concentrations of each group were analyzed. Figure 5 shows the variation in different types of soil spectral reflectance with the concentration of heavy metals in soil and the correlation coefficient between two heavy metal elements and spectral reflectance. The two heavy metals caused different changes in different soils, but they show common rules in some wavelength ranges. As in the black soil 1500–1800 nm, albic black soil 1200–2300 nm, albic soil 1000–1800 nm and meadow soil 1500–2300 nm, the spectral reflectance decreased with the increase in heavy metal concentration. The change in spectral reflectance caused by Cu is mainly concentrated in 1300–2000 nm of four soil types, and the reflectance will decrease with the increase in Cu concentration. According to the correlation curves of soil types with As and Cu concentrations, As was negatively correlated with the spectral reflectance of black soil, albic soil, and meadow soil. As was negatively correlated with the spectral reflectance of albic black soil within 500 nm, and positively correlated with it above 500 nm. Cu was negatively correlated with the spectral reflectance of four soil types. Through the correlation significance test table, *p* < 0.05 was set. According to the number of samples of each soil type, the correlation coefficient values of spectral reflectance and As and Cu concentration corresponding to the four soil types were found. The correlation coefficients of black soil, albic black soil, albic soil, and meadow soil were 0.186, 0.201, 0.182, and 0.190, respectively. The spectral bands that exceed the correlation coefficient value were selected as the characteristic bands. Table 3 shows the maximum correlation coefficient, the corresponding bands and the number of characteristic bands. Among them, As has the highest correlation with 404 nm wavelength in albic soil (−0.367); the correlation between Cu and 765 nm wavelength in black soil was the highest (−0.232).

### 3.3. Spectral Feature Selection

Redundant spectral information variables are the key factors affecting model estimation accuracy and operation efficiency. Some spectral bands in the numerous spectral bands of hyperspectral images have high autocorrelation and information redundancy degrees. Therefore, PP is employed to extract feature bands from the hyperspectral data to improve the model operation efficiency and reduce autocorrelation between spectral bands. In this study, sensitive bands were screened for the four types of soil sample data and full sample data to obtain a subset of variables with the least redundant information. During PP feature selection, the variance was selected as the projection index, the obtained projection structure of each data point of interest (the principal components) was sorted, and the contribution degree was calculated. When the cumulative contribution reached 90%, the corresponding variable was selected as the optimal variable set. Figure 6 displays the number of primary components in various soil types and entire samples that have a 90% cumulative contribution of As and Cu elements. Table 4 lists the optimal variable set and the number of variables for each soil type and for the full sample data corresponding to the two heavy metal elements. At this time, the number of characteristic bands after screening for the two heavy metal elements decreases from 145 to the 15 to 20 range, accounting for 10%~13% of the total spectral bands. This method effectively reduces dimensionality. By screening out a few important features from a large number of features, it reduces model collinearity and redundant information and computations. Moreover, the key effective information of the original data is retained.

### 3.4. Model Construction and Evaluation

To compare the model estimation results of different types of soil samples and the full sample, the sensitive bands selected by PP were used as the independent variable $X_{i}$ of the model, and the heavy metal concentration served as the dependent variable $Y_{i}$. The samples were randomly divided into a training group (model building and optimization parameters) and a validation group (model accuracy and generalization ability) according to a 3:1 ratio. LightGBM was used to perform local regression estimations of the different types of soil samples and global regression estimations of the full sample. To ascertain the effectiveness of differentiating soil types in improving the estimation accuracy and validating the estimation accuracy of soil heavy metal concentrations by the LightGBM model, the ELM and gradient-boosted decision tree (GBDT) machine learning models were used to estimate soil heavy metal concentrations. The estimation accuracy obtained from the estimation model was compared by modeling different types of soil samples and full samples, and the results are shown in Figure 7 and Table 5. The estimation accuracy of local regression was higher than that of full sample regression, according to a comparison of the three groups of estimation models. This suggests that the local regression strategy that distinguishes between soil types could effectively increase the heavy metal concentration estimation accuracy. Of these, the black soil heavy metal concentration estimation accuracy improved the most. The RPD of the three groups of models increased by 0.17 to 0.30 compared to global regression estimation, which significantly enhanced the stability of the models.

By analyzing the estimation performance of the different models, the LightGBM model outperforms the two groups of models, ELM and GBDT, in terms of prediction, obtaining higher values of $R_{P}^{2}$ and *RPD* both in the local regression and full sample regression models. The comparison results showed that the $R_{P}^{2}$ of the ELM model reached 0.60 in some soil types, while the $R_{P}^{2}$ of the GBDT model only estimating the full sample did not reach 0.60. Therefore, with limited sample data, the ensemble learning algorithm has higher accuracy than the traditional machine learning algorithms. In addition, according to the As and Cu concentration estimation results of the LightGBM model, the $R_{P}^{2}$ of different soil types models were 0.62–0.73 and 0.63–0.75, respectively. Compared with the estimation results of the other two models, the LightGBM model has a better fitting effect.

### 3.5. Soil Heavy Metal Concentration Mapping

The study area was divided into four sub-regions according to the soil type. Based on the selected characteristic spectral bands and the extracted bare soil pixels, the LightGBM model was used to draw the heavy metal concentration maps of the four sub-regions. The four sub-regions were mosaicked and merged to obtain the spatial distribution maps of soil heavy metal concentrations for the entire study area, which were compared with the spatial distribution maps of soil heavy metal concentrations drawn based on the full sample (Figure 8). According to the comparison, the spatial distributions of both regression methods showed obvious clustering characteristics, with high- and low-value areas relatively concentrated. However, the spatial distributions of heavy metal concentrations derived from local regression were more aggregated with stronger spatial heterogeneity. The estimation results for the different soils revealed that the average As concentration in the study area was 9.91 mg/kg, and the proportion between 9 mg/kg and 11 mg/kg was the highest, accounting for 64.95% of the study area. The high-value areas were mainly meadow soil and albic black soil, which accounted for 3.97% of the total area. The average Cu concentration was 21.75 mg/kg, and the high-value area accounted for 2% of the study area, with the main proportion being black soil and meadow soil. Compared to the full sample, the different soil types have more variations in the spatial distribution of heavy metal concentrations.

## 4. Discussion

### 4.1. Performance Analysis of PP–LightGBM Model Estimation

Compared with the traditional machine learning model, the PP–LightGBM model constructed by combining PP and LightGBM algorithms shows higher performance in heavy metal concentration estimation. In order to further compare the prediction ability of the PP–LightGBM model built by combining the two algorithms, LightGBM and PP-ELM were selected to establish the prediction model of two heavy metal elements on the whole sample data. The scatter diagram (Figure 9) was drawn based on the estimated and measured values of the model. The results show that the accuracy of the PP–LightGBM model is better than that of the LightGBM and PP-ELM models. The sample of training set and test set are closer to the 1:1 line, which indicates that the model has better data fitting ability. Comparing the PP–LightGBM model with the LightGBM model, it can be seen that the estimated values of the heavy metal elements estimated by the LightGBM model are relatively dispersed from the measured values. The concentration of heavy metal elements estimated by the PP–LightGBM model is more concentrated. This may be because of the noise distribution during image acquisition, which leads to more discrete estimates. The PP algorithm projects the original image data into a new feature space. In this space, the noise usually distributed in the dimension with small variance, and the signal is distributed in the dimension with large variance. Therefore, the degree of the model affected by noise can be reduced by eliminating some noise information, and the accuracy of its estimation can be improved. On the other hand, the comparison between the PP–LightGBM and PP-ELM models shows that the estimation of heavy metal concentration by the PP-ELM model is not accurate enough, the estimation of As is generally high, and the estimation of Cu is generally low. The predicted and true values of heavy metal elements obtained by the PP–LightGBM model are more accurate, and the training set and verification set are close to the 1:1 line. Moreover, the validation set $R^{2}$ estimated by PP–LightGBM is significantly higher than that estimated by PP-ELM. This may be because, as a Boosting family iterative algorithm, LightGBM will adjust the weight of the sample in each iteration according to the estimated results of the previous iteration. Therefore, with the progress of iteration, the error will gradually decrease, the deviation of the model will continue to decrease, and the final accuracy will be improved. This makes the LightGBM algorithm more suitable for processing large-scale data sets, which is consistent with the research results of McCarty et al. [53].

### 4.2. Analysis of Heavy Metal Estimation Results in Different Soils

According to the comparison between the accuracy of heavy metal concentrations in the whole sample and different soil type samples (Figure 10), the local modeling considering the soil type differences effectively improved the model estimation accuracy. The reasons are as follows. For one thing, the structures of different soils are different due to soil moisture changes, tillage, freeze–thaw cycles, and the physical pressures of expansion and contraction induced by the movement of larger biological populations in the soil [54]. Black soil has a loose structure arranged in aggregates or blocks, which helps maintain soil aeration and drainage. The albic soil and meadow soil structures are mainly blocky or columnar, leading to relatively poor drainage. Sample collection in this study was conducted under dry soil conditions. However, the physical structures of different soils differ, and consequently, their intrinsic moisture concentration varies, which may affect the estimation of heavy metal concentrations with full sample data and lead to lower estimation accuracy. Among them, black soil is less affected by water because of its good soil structure, and the accuracy is higher when estimating the heavy metal concentration in local modeling. For another, different types of soils have different chemical components. In addition to the large differences in physicochemical properties between soils, the concentrations of different soil components [55] will also lead to different soils having different heavy metal adsorption capacities and, thus, lower estimation accuracy. Iron-manganese oxides in soil have a strong adsorption effect on As [56], and iron-manganese nodules, as aggregates of iron-manganese oxides, have the same properties as iron-manganese oxides. Although the soil formation processes of meadow soil and albic black soil differ, they contain large amounts of iron-manganese nodules and iron-manganese sediments and also have strong adsorption capacities for As. The visible–near-infrared reflectance spectrum showed a strong connection with the Cu concentration, which was mainly affected by organic matter [57]. Black soil and meadow soil have abundant humus layers and high organic matter accumulations, and their Cu adsorption capacity is also relatively strong. Therefore, by considering the soil type differences, conducting local regression-based soil heavy metal concentration estimation can effectively reduce the influence due to soil structure and component differences, thus improving the model estimation accuracy.

### 4.3. Uncertainty Analysis of Heavy Metal Concentration Estimation for Different Soils

The spatial distribution results obtained by inversion of heavy metal concentration in the study area based on the PP–LightGBM model show that the concentration range of the two elements is basically consistent with the research results of Peng et al. [58]. However, the concentration of some areas is higher than the background value. Figure 11 reveals the spatial distribution and topographic data analysis results of heavy metals in the study area. It can be seen that although there is no industry in the study area, the production activities in residential areas may be one of the main reasons for the accumulation of heavy metals. The high concentration of heavy metals in the surrounding soil may be related to the accumulation of domestic waste, accumulation of fertilizer, and sewage treatment and transportation [59]. In addition, due to the small size of the residential area and the limited environmental impact of the activities, the analysis was combined with topographic data. The results show that there is a significant relationship between the topographic fluctuation and the accumulation and diffusion of heavy metals. The topography in the study area is mostly hilly and undulating, and the heavy metals mainly accumulate in low-lying farmland. This may be because of the increased surface runoff caused by rainy and snowy weather conditions, which makes the water accumulate in the relief of the topography and carry away pollutants from residential areas, and finally accumulate in farmland.

In the analysis of satellite remote sensing images, the spectral information is not only affected by terrain factors, also by humidity, roughness, and ground vegetation [60]. In addition, weather and topographic differences also affect the spectral reflectance during image acquisition, and they may affect the estimation accuracy of soil heavy metal concentration. To ensure the reliability of the data, we chose the appropriate satellite transit time and weather conditions for sampling, and ensured that the sampling points were located in pure soil pixels to reduce the impact of spectral information. Other factors affecting the soil spectral reflectance include the parent material, mineral composition, surface roughness, particle size, and water content [61]. Soil moisture has a great influence on the spectral reflectance of visible light and near infrared light. In subsequent research, establishing the relationship between the spectrum and the pure soil based on the ground measured data will help to improve the prediction accuracy of the model. Meanwhile, due to the differences in physical and chemical properties, different types of soil may cause differences in hyperspectral characteristics. Therefore, whether the establishment of a unified model for soils with large differences in hyperspectral characteristics will have a greater impact on the prediction accuracy of the model requires deeper research to verify.

## 5. Conclusions

In this paper, a PP–LightGBM model was constructed by using PP and the LightGBM algorithm and a method of estimating heavy metal concentration in soil using hyperspectral images was proposed and its feasibility was discussed. The accuracy and stability of the model were effectively improved by local regression estimation. After further analysis, it was found that the PP algorithm has an advantage in the face of information redundancy and high correlation between hyperspectral bands and successfully optimized the projection of high-dimensional data into low-dimensional space for calculation, reducing the computational complexity and contributing to improving the accuracy of subsequent model estimation. In terms of model evaluation, the LightGBM algorithm shows better data fitting ability compared with traditional machine learning models. The PP–LightGBM model, which combined the two algorithms, has good accuracy and generalization in the estimation of soil heavy metal concentration and has obtained a good estimation effect. In addition, considering the differences in physical and chemical properties and spectral characteristics of different soils, establishing local regression for different soil types can reduce the impact of these differences on the model estimation and improve the estimation accuracy. The study also found that the distribution of heavy metal accumulation is affected by many factors such as human activities and topographic conditions. Therefore, in the evaluation of soil environmental quality and agricultural ecological protection, special attention should be paid to the accumulation of heavy metals in agricultural land adjacent to residential areas and surrounding low-lying topographies. In summary, the results of this study can provide theoretical support for the comprehensive treatment of soil heavy metals and provide an important reference for the improvement of the agricultural ecological environment.

## Figures and Tables

**Figure 1 sensors-24-03251-f001:**
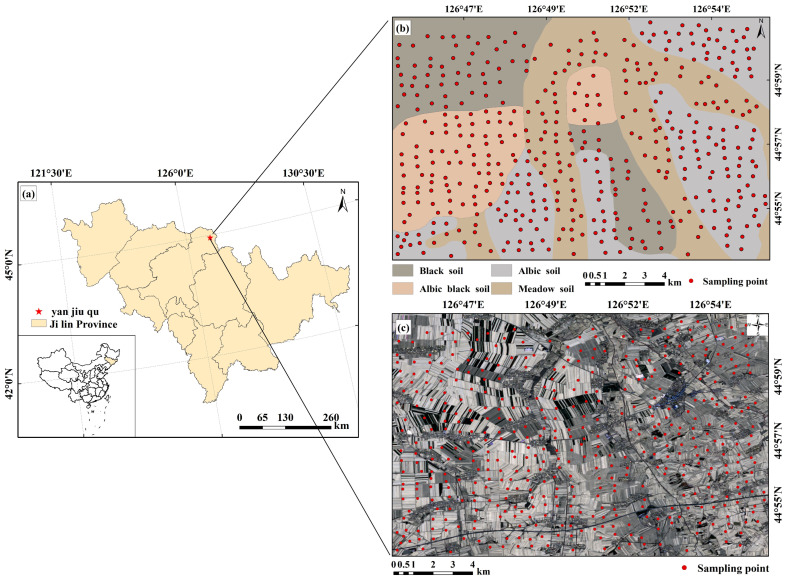
Map of the study area. (**a**) Sihe Town and its surrounding farming area in Jilin Province, China; (**b**) sampling site distribution map in the study area; (**c**) soil type distribution map in the study area.

**Figure 2 sensors-24-03251-f002:**
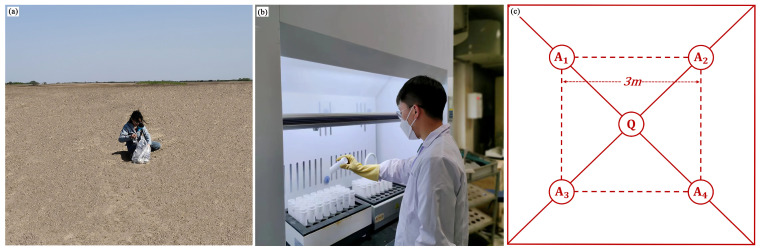
Soil sample pretreatment. (**a**) Sample plot diagram of soil sample collection; (**b**) soil sample collection diagram; (**c**) heavy metal concentration determination chart for the collected samples.

**Figure 3 sensors-24-03251-f003:**
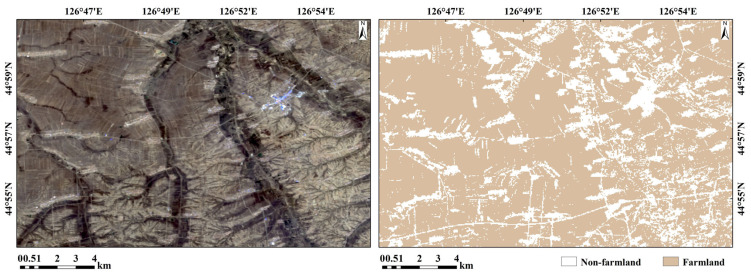
Random forest-supervised classification extraction of bare soil image element results.

**Figure 4 sensors-24-03251-f004:**
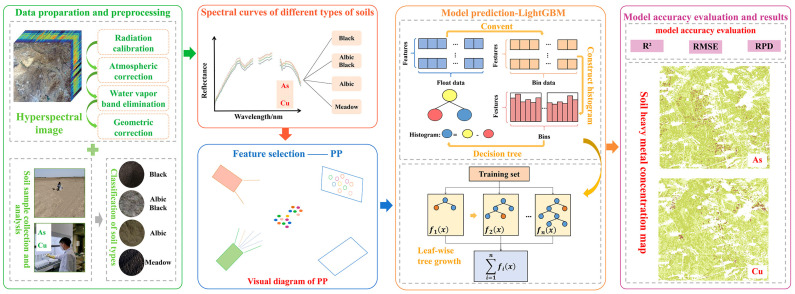
Flow chart of hyperspectral image-based heavy metal concentration estimation in various soil types.

**Figure 5 sensors-24-03251-f005:**
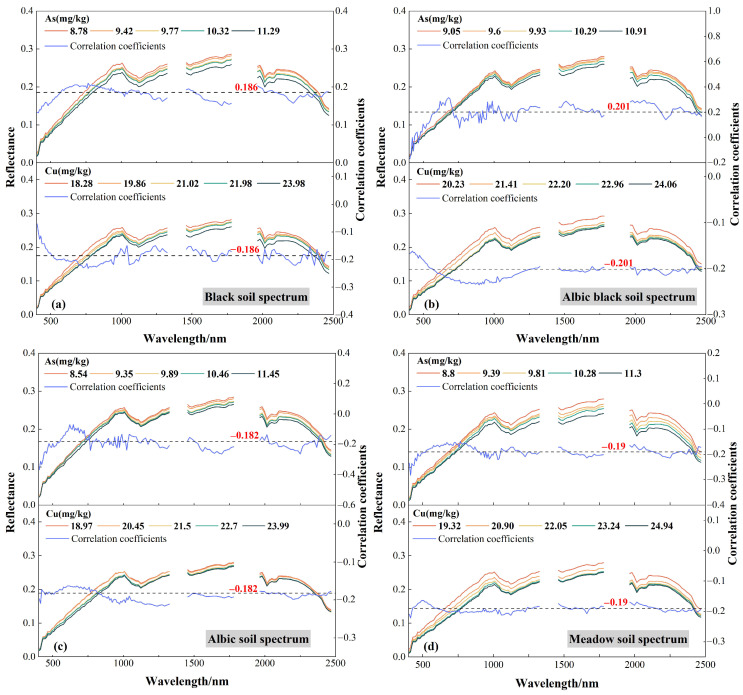
Spectral curves of different soils and correlation with heavy metal elements. (**a**) Black soil spectral curve; (**b**) Albic black soil spectral curve; (**c**) Albic soil spectral curve; (**d**) Meadow soil spectral curve.

**Figure 6 sensors-24-03251-f006:**
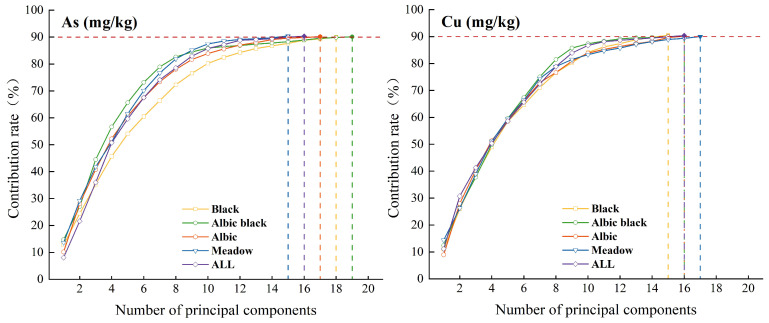
The cumulative contribution rate of the feature information’s major components. Yellow dotted line is Black soil; green dotted line is Albic black soil; red dotted line is Albic soil; blue dotted line is Meadow soil; purple dotted line is All soil.

**Figure 7 sensors-24-03251-f007:**
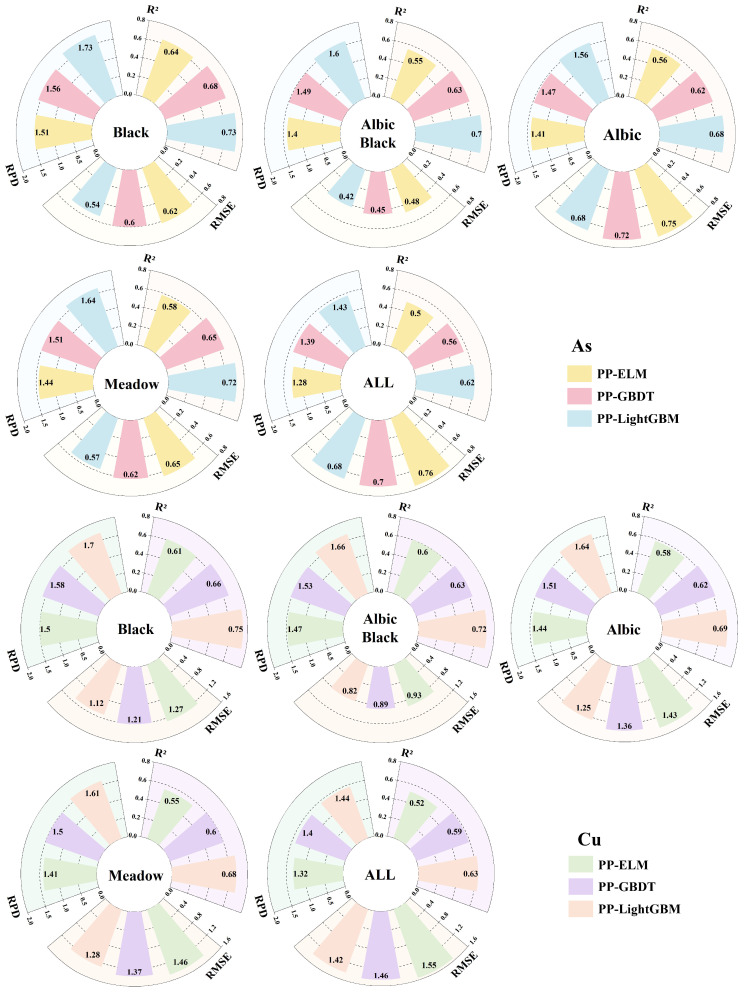
A comparison of heavy metal concentration estimation accuracy based on different soil types.

**Figure 8 sensors-24-03251-f008:**
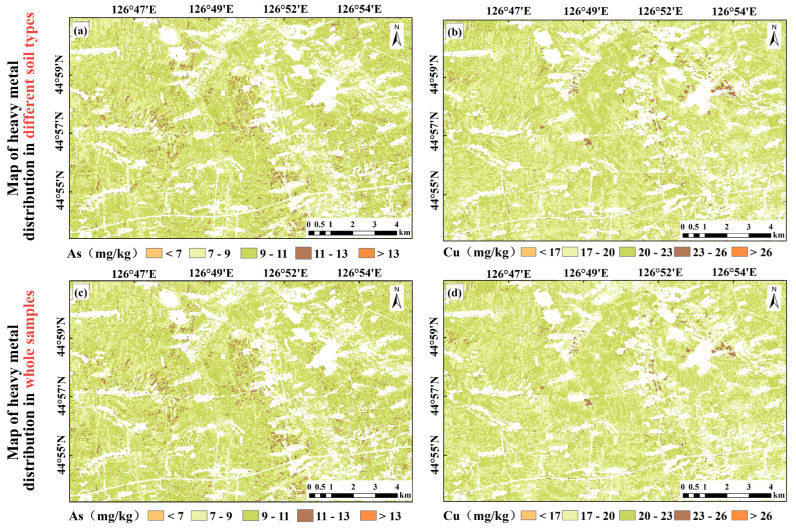
Mapping of soil heavy metal concentration for the study area. (**a**,**b**) Map of heavy metal distribution in different soil types; (**c**,**d**) map of heavy metal distribution in whole samples.

**Figure 9 sensors-24-03251-f009:**
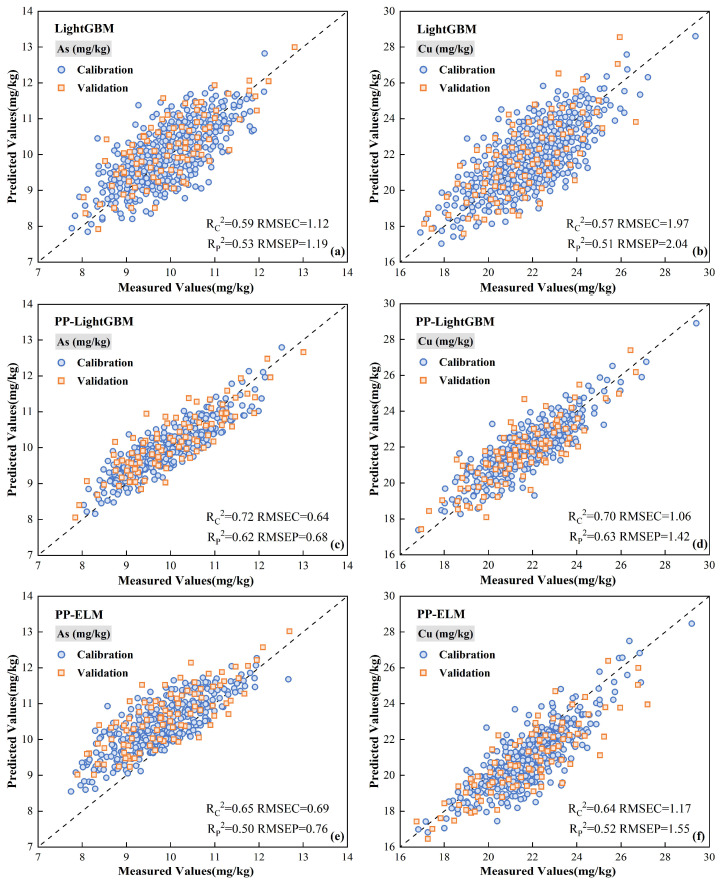
Measured and predicted values. (**a**) LightGBM of soil As concentration; (**b**) LightGBM of soil Cu concentration; (**c**) PP–LightGBM of soil As concentration; (**d**) PP–LightGBM of soil Cu concentration; (**e**) PP-ELM of soil As concentration; (**f**) PP-ELM of soil Cu concentration.

**Figure 10 sensors-24-03251-f010:**
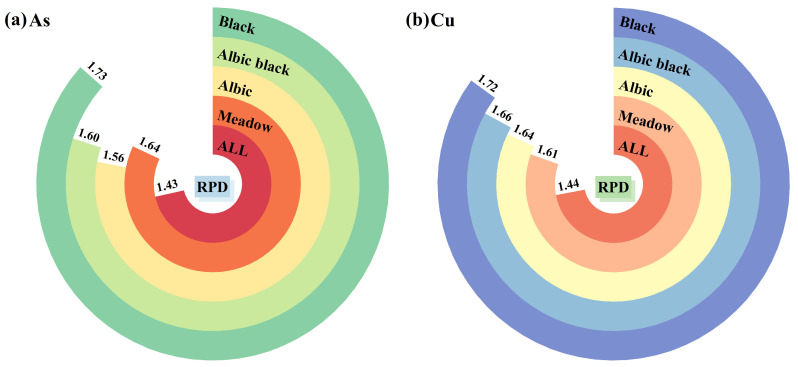
Estimation accuracy (*RPD*) of the PP–LightGBM soil heavy metal concentration model. (**a**) Estimation accuracy (*RPD*) of As by PP-LightGBM model; (**b**) Estimation accuracy (*RPD*) of Cu by PP-LightGBM model.

**Figure 11 sensors-24-03251-f011:**
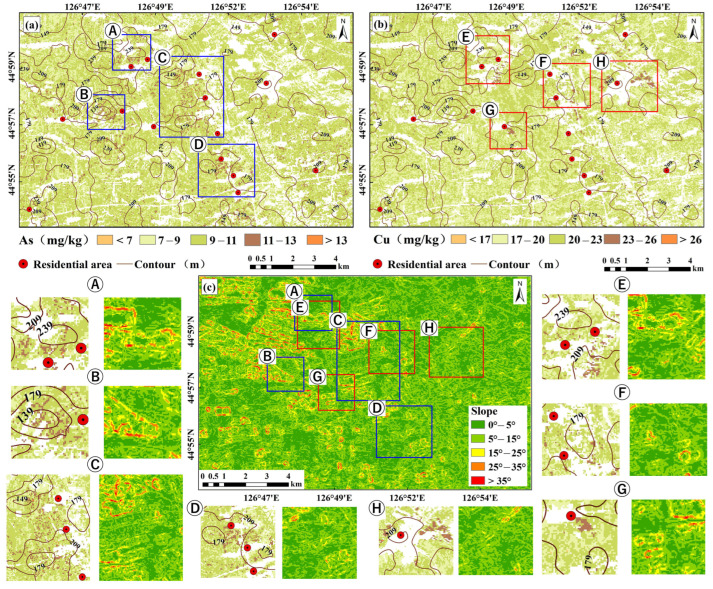
Spatial distribution of heavy metal concentration and topographic data overlay analysis. (**a**) As superposition analysis results; (**b**) Cu superposition analysis results; (**c**) slope calculation results of the study area; A–D represents the area where As elements gather; E–H represents the area where Cu elements gather.

**Table 1 sensors-24-03251-t001:** The main training parameters of the LightGBM algorithm.

Parameters	Value	Description
num_leaves	31	The most leaves that a tree can have
learning_rate	0.05	Improves learning rate
early_stopping_rounds	50	If the model loss does not drop in the specified rounds, the training will be stopped.

**Table 2 sensors-24-03251-t002:** Heavy metal concentration information of the soil samples (Unit: mg/kg).

Element	Type	Max	Min	Mean	Std	CV	Background ^1^	National ^2^
As	Black	11.63	8.16	9.93	0.67	0.067	6.70	15.00
	Albic Black	13.14	7.24	9.95	0.93	0.094		
	Albic	12.82	7.32	9.94	1.06	0.106		
	Meadow	13.53	7.35	9.96	0.94	0.094		
Cu	Black	29.51	17.52	22.17	1.91	0.144	16.40	35.00
	Albic Black	26.86	16.76	21.56	2.06	0.098		
	Albic	25.37	19.14	21.05	1.36	0.061		
	Meadow	27.22	16.51	22.13	2.06	0.093		

^1^ China Environmental Monitoring Station. Background Value of Soil Elements in China. China Environmental Science Press. ^2^ National Environmental Protection Agency. Environmental quality standard for soils Beijing.

**Table 3 sensors-24-03251-t003:** Maximum correlation coefficient and feature bands.

Type		As	Cu
Black	Maximum correlation coefficient	0.209	−0.232
	Corresponding band	765	765
	Number of sensitive bands	65	81
Albic black	Maximum correlation coefficient	0.317	−0.226
	Corresponding band	679	894
	Number of sensitive bands	75	94
Albic	Maximum correlation coefficient	−0.367	−0.220
	Corresponding band	404	1031
	Number of sensitive bands	84	92
Meadow	Maximum correlation coefficient	−0.282	−0.222
	Corresponding band	413	413
	Number of sensitive bands	65	91

**Table 4 sensors-24-03251-t004:** PP feature extraction results.

Elements	Type	Wavelength (nm)	Number of Principal Components	Cumulative Contribution (%)
As	Black	885, 859, 988, 2048, 1040, 542, 765, 1543, 2317, 550, 1072, 413, 2400, 507, 1031, 1274, 524, 816	18	90.00
	Albic Black	481, 1223, 1660, 1475, 593, 954, 885, 2014, 585, 490, 1728, 1173, 413, 2233, 679, 1526, 507, 455, 1745	19	90.11
	Albic	1963, 1492, 902, 404, 619, 705, 1576, 2266, 2132, 894, 499, 2283, 1207, 945, 713, 602, 516	17	90.18
	Meadow	696, 1274, 2148, 1745, 1677, 550, 2115, 636, 1526, 413, 1240, 2484, 2199, 516, 619	15	90.44
	ALL	679, 1778, 885, 404, 662, 455, 2334, 2081, 765, 1459, 868, 1492, 1257, 610, 1023, 413	16	90.33
Cu	Black	928, 1963, 2334, 473, 851, 1031, 765, 499, 645, 2199, 1980, 2434, 1223, 1190, 653	15	90.50
	Albic Black	2216, 774, 894, 490, 481, 902, 713, 1190, 876, 421, 2014, 653, 662, 687, 413, 1173	16	90.14
	Albic	542, 756, 1031, 842, 1475, 1644, 679, 1190, 791, 507, 1778, 2233, 2182, 765, 653, 2081	16	90.17
	Meadow	516, 945, 2417, 1207, 1745, 1526, 705, 619, 876, 413, 808, 1023, 499, 971, 1105, 2014, 774	17	90.03
	ALL	928, 670, 524, 765, 2366, 1324, 1627, 413, 1711, 894, 1610, 1728, 1644, 2199, 902, 937	16	90.37

**Table 5 sensors-24-03251-t005:** Accuracy metrics of model regression results.

		As			Cu		
Type	Model	$R_{P}^{2}$	${RMSE}_{P}$	RPD	$R_{P}^{2}$	${RMSE}_{P}$	RPD
Black	PP-ELM	0.64	0.62	1.51	0.61	1.27	1.50
	PP-GBDT	0.68	0.60	1.56	0.66	1.21	1.58
	PP–LightGBM	0.73	0.54	1.73	0.75	1.12	1.72
Albic Black	PP-ELM	0.55	0.48	1.40	0.60	0.93	1.47
	PP-GBDT	0.63	0.45	1.49	0.63	0.89	1.53
	PP–LightGBM	0.70	0.42	1.60	0.72	0.82	1.66
Albic	PP-ELM	0.56	0.75	1.41	0.58	1.43	1.44
	PP-GBDT	0.62	0.72	1.47	0.62	1.36	1.51
	PP–LightGBM	0.68	0.68	1.56	0.69	1.25	1.64
Meadow	PP-ELM	0.58	0.65	1.44	0.55	1.46	1.41
	PP-GBDT	0.65	0.62	1.51	0.60	1.37	1.50
	PP–LightGBM	0.72	0.57	1.64	0.68	1.28	1.61
All	PP-ELM	0.50	0.76	1.28	0.52	1.55	1.32
	PP-GBDT	0.56	0.70	1.39	0.59	1.46	1.40
	PP–LightGBM	0.62	0.68	1.43	0.63	1.42	1.44

## Data Availability

The data presented in this study are available on request from the corresponding author.

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
