# Peer review of "Heavy Metal Concentration Estimation for Different Farmland Soils Based on Projection Pursuit and LightGBM with Hyperspectral Images"

_sensors, 2024, doi:10.3390/s24103251_

Round 1

Reviewer 1 Report

Comments and Suggestions for Authors

Detecting heavy metal concentration in agriculture lands using advanced sensing technologies is interesting topic and within the scope of the journal. However, a major revision is require. Below are my comments.

The authors should differentiate between “Heavy metal content” and “concentration”. I think you are measuring the concentration of the heavy metals not the content. Please check your datasets and the procedure of collecting your samples, and then revise the text and figures as well as the title of the manuscript. 

Lines 16-23. The introductory part in the abstract is too long (about 8 lines) please shorten into 2-3 lines.

Line 132: “ZY1-02D remote sensing images were used as the data source” please add one sentence about this source of data.

140-145: The objectives need to be revised. The language is not clear at all.  

I don’t understand why you select the soil to be your target but not plants. In heavy metal studies, our concern is the plant more than the soil.

Also why you select As an Cu metals. The most heavy metals that we are concern about are Pb Cd and Cr. The reasons for selecting Cu and As should be included. Are the agricultural land in your study area suffer from those elements? You need to support your answers with historical data or references. 

Lines 170-173: I think this is the main drawback of this study. To apply this model or approach in the future, you need to make sure that the area (100-200 m) is clean, no vegetation, no stones (only bare soil) in order to reduce the impact of the image-mixing pixels. In my opinion, sending the samples to the laboratory will be cheaper than cleaning the area from grasses and stones.  

Line 205: what are the FLAASH module inputs?

Lines 235-248: “Comprehensive pollution index” I don’t think your dataset are adequate to calculate this index. In fact to assess the contamination in the soil by heavy metals we need to get soil samples multiple times across two-three growing seasons. Saying that the soil is contaminated by heavy metal can lead to potential risk to the farmers in these lands (no one will have there products in the future). Therefore, the assessment process should be fair, clear and scientific.  

320-322: “The Nemerow comprehensive pollution indices of Cu for black soil and meadow soil reached the warning threshold, and the Nemerow comprehensive pollution indices for the remaining two types of soils were within the safe range.”  You cannot say this based on one sampling date.

Table 2: what does “National’ mean?

Figure 5 shows the spectral curves of different soils. However, no statistical analysis has been made across the spectral bands. Without statistical analysis for at least few bands, this figure does not add any scientific (accurate) information.   

Lines 332-339: The authors should use the words ‘high’ ‘higher’ ‘significant’ ‘lower’ only if the means or treatments are statistically different. Figure 5 had no statistical analysis, therefore, all the text related to this figure should be revised.

Lines 494-498: How we can guarantee these inputs or micro-environment if we plan to use this approach (model) in the future in another site.  

Discussion part is very poor. No in-depth assessment and discussion have been made. The main goal of this study is to test the accuracy of the remotely-sensed model. The authors did not clearly explain why the model was not perfect, or highly accurate. How to improve the model accuracy in the future. What are the drawbacks of this model etc.. In fact, about 40%-50% of the discussion was about soil contamination and why it’s happen. No solid data in this study can approve this point of view. I cannot say the soil is in danger or contaminated based on one sampling date.

Author Response

Responds letter

Thank you for your letter and for the reviewers’ comments concerning our manuscript entitled “Heavy metal concentration estimation for different farmland soils based on projection pursuit and LightGBM with hyper-spectral images”. Those comments are all valuable and very helpful for revising and improving our paper, as well as the important guiding significance to our researches. We have studied comments carefully and have made correction which we hope meet with approval. The main corrections in the paper and point-by-point response to the reviewer’s comments are as flowing:

Responds to reviewer1s comments:

Point 1: The authors should differentiate between “Heavy metal content” and “concentration”. I think you are measuring the concentration of the heavy metals not the content. Please check your datasets and the procedure of collecting your samples, and then revise the text and figures as well as the title of the manuscript. 

Response:

Thanks for the reviewer’s suggestion. According to your suggestions, we checked the text of the article. In order to reflect the main research more accurately, we have corrected the “heavy metal content” in the text and chart to “heavy metal concentration”.

Point 2: Lines 16-23. The introductory part in the abstract is too long (about 8 lines) please shorten into 2-3 lines.

Response:

Thanks for the reviewer’s suggestion. According to your suggestions, we refine and summarize the parts introduced in the abstract (lines 16-19) and ensures the accuracy of the overall narrative. It improves the quality and readability of the abstract.

Point 3: Line 132: “ZY1-02D remote sensing images were used as the data source” please add one sentence about this source of data.

Response:

Thanks for the reviewer’s suggestion. According to your suggestions, we have adjusted the description of "ZY1-02D remote sensing images were used as the data source" in the article 135 line to the 130 line so that the reader can understand it better.

Point 4: 140-145: The objectives need to be revised. The language is not clear at all.  

Response:

Thanks for the reviewer’s suggestion. According to your suggestions, we have rearranged the research objectives (lines 142-147) according to the main research of the paper. This better reflects the direction of our research.

Point 5: I don’t understand why you select the soil to be your target but not plants. In heavy metal studies, our concern is the plant more than the soil.

Response:

Thank you very much for your suggestion. For your suggestion, I would like to make some additional remarks: This study is based on hyperspectral imaging technology, which uses land parcel samples (image pixels) as research objects. However, due to the limitation of image pixels (30m×30m), it is impossible to ensure that the number and species of plants in each pixel are consistent. Therefore, the use of plants as research objects may not be fully informative. In contrast, bare soil pixels have suitable conditions for the use of hyperspectral imaging techniques. This is because bare soil areas are usually not occluded by vegetation and have fewer interfering factors. Therefore, it may be more appropriate to use bare soil as the research object to effectively utilize the information provided by hyperspectral imaging technology.

Point 6: Also why you select As an Cu metals. The most heavy metals that we are concern about are Pb Cd and Cr. The reasons for selecting Cu and As should be included. Are the agricultural land in your study area suffer from those elements? You need to support your answers with historical data or references. 

Response:

Thank you very much for your suggestion. For your suggestion, I would like to make some additional remarks: the selection of monitoring elements in the study area is based on geochemical census data. These two elements show a tendency to be higher than the background value in the results, so this paper chooses these two heavy metal elements. According to your suggestion, we have added a description of the historical data in the text 55-57 lines.

Point 7: Lines 170-173: I think this is the main drawback of this study. To apply this model or approach in the future, you need to make sure that the area (100-200 m) is clean, no vegetation, no stones (only bare soil) in order to reduce the impact of the image-mixing pixels. In my opinion, sending the samples to the laboratory will be cheaper than cleaning the area from grasses and stones.  

Response:

Thank you very much for your suggestion. For your suggestion, I would like to make some additional remarks: the locations mentioned in this study, which are more than 100 m apart from the surrounding ground objects and only have bare soil, are the sampling points for the laboratory analysis of heavy metal concentration (lines 172-176). In addition, in order to minimize the influence of mixed pixels in the image, this study selected sampling in the bare soil period (crop cultivation period) and hyperspectral image acquisition.

Point 8: Line 205: what are the FLAASH module inputs?

Response:

Thanks for the reviewer’s suggestion. According to your suggestions, we provide a supplementary account in which we used the FLAASH model in the ENVI software for atmospheric correction of the images to eliminate radiometric errors caused by atmospheric molecular and aerosol scattering (lines 208-211).

Point 9: Lines 235-248: “Comprehensive pollution index” I don’t think your dataset are adequate to calculate this index. In fact to assess the contamination in the soil by heavy metals we need to get soil samples multiple times across two-three growing seasons. Saying that the soil is contaminated by heavy metal can lead to potential risk to the farmers in these lands (no one will have there products in the future). Therefore, the assessment process should be fair, clear and scientific.  

Response:

Thanks to the reviewer’s suggestion, since we have described the growth trend of these two heavy metal elements in the study area in the text 55-57 lines, we have deleted the description and results of the comprehensive pollution index in the text and Table 2 in this part.

Point 10: 320-322: “The Nemerow comprehensive pollution indices of Cu for black soil and meadow soil reached the warning threshold, and the Nemerow comprehensive pollution indices for the remaining two types of soils were within the safe range.”  You cannot say this based on one sampling date.

Response:

Thanks to the reviewer’s suggestion, based on this and the previous suggestion, we have deleted this part and the text description to ensure the rationality of the article.

Point 11: Table 2: what does “National’ mean?

Response:

Thanks for the reviewer’s suggestion. According to your suggestions, we supplement this in the comments below Table 2. "National" in Table 2 refers to the background values of soil elements in various regions of China published by China Environmental monitoring stations.

Point 12: Figure 5 shows the spectral curves of different soils. However, no statistical analysis has been made across the spectral bands. Without statistical analysis for at least few bands, this figure does not add any scientific (accurate) information.   

Response:

Thanks for the reviewer's suggestion. According to your suggestion, we have added the statistical analysis of spectral bands in lines 338-351 and Figure 5. Including the correlation between soil spectral and heavy metal concentrations and the number of characteristic bands screened according to the correlation.

Point 13: Lines 332-339: The authors should use the words ‘high’ ‘higher’ ‘significant’ ‘lower’ only if the means or treatments are statistically different. Figure 5 had no statistical analysis, therefore, all the text related to this figure should be revised.

Response:

Thanks for the reviewer’s suggestion. According to your suggestions, we have corrected the words “high”, “higher”, “significant” and “lower”, and made statistical analysis on Figure 5 (lines 333-351).

Point 14: Lines 494-498: How we can guarantee these inputs or micro-environment if we plan to use this approach (model) in the future in another site.  

Response:

Thanks for the reviewer’s suggestion. According to your suggestion, we have supplemented in the discussion of section 4.3. In future research, we will focus on improving the model applicability in different scenarios. Specifically, we will conduct in-depth research on spectral correction and soil condition differences, such as introducing environmental variables to improve the estimation accuracy of the model. In addition, we will explore and test the models with transfer learning capabilities to further optimize their performance.

Point 15: Discussion part is very poor. No in-depth assessment and discussion have been made. The main goal of this study is to test the accuracy of the remotely-sensed model. The authors did not clearly explain why the model was not perfect, or highly accurate. How to improve the model accuracy in the future. What are the drawbacks of this model etc.. In fact, about 40%-50% of the discussion was about soil contamination and why it’s happen. No solid data in this study can approve this point of view. I cannot say the soil is in danger or contaminated based on one sampling date.

Response:

Thanks for the reviewer’s suggestion. According to your suggestions, we have revised the article Discussion 4.1 to include a description of the advantages of the PP-LightGBM model and ways in which the model accuracy can be improved in future research.

Reviewer 2 Report

Comments and Suggestions for Authors

Critical response on manuscript entitled “Heavy metal content estimation for different farmland soils based on projection pursuit and LightGBM with hyperspectral images” written by Nan Lin and others and submitted to Sensors

The manuscript under consideration has shown impressive research results. Developed paper is dedicated to the applications of remote sensing with hyperspectral images and machine learning (ML) for detecting of presence and distribution of pollutants (Cu and As) in farmlands. Authors convincingly shown  that projection pursuit – LightGBM model is applicablfor building a local pollutant distribution models and proved that with assessing metrics. Paper has 60 references on the mostly peer-reviewed sources. Authors also tend to achieve decreasing of redundant dimensionality ofhyperspectral dataThe characteristics of ZY1-02D satellite are given. Pixel digital numbers (DN) data from images were converted to reflectance.

Also, authors use considerable dataset of field observation points to track As and Cu distribution in soils and determine the soil type.

However, some details needs to be clarified to make the paper valuable for the international reader, as shown below:

In a text form of presence (ion, mineral) of pollutant elements so they can affect spectral quantities of the surface isn’t clarified. 

Make sure that all abbreviations are expanded in the text before initial use. 

Figure 4. Fonts and sub-pictures aren’t readable. Internationally acceptable ISO9001 standard charts can shows dataflow better. Figure requires simplification.

Table 1. How main parameters were determined? Based on which conventional metrics? 

Table 2. Abbreviations must be expanded in the text or table description.

Figure 7. Accuracy metrics is hard to compare basically on the pictureworth trying to summarize it in a table. 

Figure 8 is not readable. Coordinate network and class names is hard to read and understand. No scale bars! 

Figure 9. Hard to read and understand isolines, no scale bars are shown. At such scale it is better to use markers to pinpoint the anomalies.

Line 549. Data sharing is DEFINETELY applicable here, specifically in terms of scaled (standartized or normalized geochemical data),  LightGBM model configuration to support reproducivity of the results. 

Conclusion. It is hard to see, to what extent LGBM modeling and hyperspectral data inclusion facilitated mapping of pollutant distribution. Obviously, satellite data could either decrease amount of field sampling or provide a model transferrable to another area with similar soil 

composition. Are any of mentioned models applied is not clear. Based on that I recommend major revision

Author Response

Responds letter

Thank you for your letter and for the reviewers’ comments concerning our manuscript entitled “Heavy metal concentration estimation for different farmland soils based on projection pursuit and LightGBM with hyper-spectral images”. Those comments are all valuable and very helpful for revising and improving our paper, as well as the important guiding significance to our researches. We have studied comments carefully and have made correction which we hope meet with approval. The main corrections in the paper and point-by-point response to the reviewer’s comments are as flowing:

Responds to reviewer2s comments:

Point 1: In a text form of presence (ion, mineral) of pollutant elements so they can affect spectral quantities of the surface isn’t clarified. 

Response:

Thanks for the reviewer’s suggestion. According to your suggestion, we have added a supplementary analysis on the correlation between heavy metal elements and soil spectra in lines 319-322 of the paper.

Point 2: Make sure that all abbreviations are expanded in the text before initial use. 

Response:

Thanks for the reviewer’s suggestion. I'm really sorry. According to your suggestions, we went through all the abbreviations and corrected the initial use of an undisplayed abbreviation.

Point 3: Figure 4. Fonts and sub-pictures aren’t readable. Internationally acceptable ISO9001 standard charts can shows dataflow better. Figure requires simplification.

Response:

Thanks for the reviewer’s suggestion. According to your suggestions, we further combed and summarized Figure 4. And presented it in a clearer way.

Point 4: Table 1.How main parameters were determined? Based on which conventional metrics? 

Response:

Thanks for the reviewer’s suggestion. According to your suggestions, We supplement the determination of the main parameters of the LightGBM model in lines 278-285.

Point 5: Table 2. Abbreviations must be expanded in the text or table description.

Response:

Thanks for the reviewer’s suggestion. I'm really sorry. According to your suggestions, we expanded the text and table description abbreviations in Table 2.

Point 6: Figure 7. Accuracy metrics is hard to compare basically on the picture, worth trying to summarize it in a table. 

Response:

Thanks for the reviewer’s suggestion. According to your suggestions, We supplement Figure7 with a table (Table5).

Point 7: Figure 8 is not readable. Coordinate network and class names is hard to read and understand. No scale bars! 

Response:

Thanks for the reviewer’s suggestion. According to your suggestions, we have modified Figure 8, and the coordinate grid in the modified Figure 8 has been enlarged, the class names have been re-described, the "different soil types" and "full samples" have been distinguished, and the scale has been added to the figure for readers to read more clearly.

Point 8: Figure 9. Hard to read and understand isolines, no scale bars are shown. At such scale it is better to use markers to pinpoint the anomalies.

Response:

Thanks for the reviewer’s suggestion. According to your suggestions, we have modified Figure 9, and the contour lines in the modified Figure 9 (Now Figure 11) are positioned in a marked way, and the scale is added in the figure for readers to read more clearly.

Point 9: Line 549. Data sharing is DEFINETELY applicable here, specifically in terms of scaled (standartized or normalized geochemical data),  LightGBM model configuration to support reproducivity of the results. 

Response:

Thanks for the reviewer’s suggestion. However, I would like to explain that this research is funded by the National Natural Science Foundation of China and is still in the confidential stage. Due to the confidentiality of the research content, we cannot provide data sharing. Thank you for your understanding.

Point 10: Conclusion. It is hard to see, to what extent LGBM modeling and hyperspectral data inclusion facilitated mapping of pollutant distribution. Obviously, satellite data could either decrease amount of field sampling or provide a model transferrable to another area with similar soil composition. Are any of mentioned models applied is not clear.

Response:

Thanks for the reviewer’s suggestion. According to your suggestions, we provide a further summary of the conclusions. For this I would like to explain: we further summarize and refine our research results in the conclusion based on the research content. In terms of model construction, the PP and the LightGBM algorithm have complementarity in processing data. In this paper, it is verified that PP-LightGBM has higher accuracy and better estimation performance than traditional models. At the same time, based on the analysis of the spatial distribution map, the reliability of retrieving heavy metal concentration based on hyperspectral imaging technology is explained.

Round 2

Reviewer 1 Report

Comments and Suggestions for Authors

The authors responded properly to my comments 

Reviewer 2 Report

Comments and Suggestions for Authors

Dear Authors, 

Thanks for comprehensive responses for comments. 

The manuscript in present form is appropriate for publication.

Regards, Reviewer